

# Machine learning of symbolic compositional rules with genetic programming: dissonance treatment in Palestrina

Torsten Anders[1] and Benjamin Inden[2]

[1] School of Media Arts and Performance, University of Bedfordshire, Luton, Bedfordshire, UK
[2] Department of Computer Science and Technology, Nottingham Trent University, Nottingham, UK

## ABSTRACT

We describe a method for automatically extracting symbolic compositional rules from music corpora. Resulting rules are expressed by a combination of logic and numeric relations, and they can therefore be studied by humans. These rules can also be used for algorithmic composition, where they can be combined with each other and with manually programmed rules. We chose genetic programming (GP) as our machine learning technique, because it is capable of learning formulas consisting of both logic and numeric relations. GP was never used for this purpose to our knowledge. We therefore investigate a well understood case in this study: dissonance treatment in Palestrina's music. We label dissonances with a custom algorithm, automatically cluster melodic fragments with labelled dissonances into different dissonance categories (passing tone, suspension etc.) with the DBSCAN algorithm, and then learn rules describing the dissonance treatment of each category with GP. Learning is based on the requirement that rules must be broad enough to cover positive examples, but narrow enough to exclude negative examples. Dissonances from a given category are used as positive examples, while dissonances from other categories, melodic fragments without dissonances, purely random melodic fragments, and slight random transformations of positive examples, are used as negative examples.

## INTRODUCTION

Artificial intelligence methods have been used for decades to model music composition (*Fernández & Vico, 2013*). Two general approaches have attracted particular attention, as they mimic two aspects of how humans learn to compose. Firstly, rules have been used for centuries for teaching composition. Some algorithmic composition methods model symbolic knowledge and rules, for example, constraint-based approaches and formal grammars. Secondly, composers learn from examples of existing music. Machine learning (ML) methods of algorithmic composition include Markov chains, and artificial neural networks.

Corresponding author
Benjamin Inden,
benjamin.inden@ntu.ac.uk

We aim at combining these two approaches by automatically learning compositional rules from music corpora. We use genetic programming (GP) (*Poli, Langdon & McPhee, 2008*) for that purpose.

The resulting rules are represented symbolically, and can thus be studied by humans (in contrast to, say, artificial neural networks), but the rules can also be integrated into algorithmic composition systems. Extracting rules automatically is useful, for example, for musicologists to better understand the style of certain corpora, and for composers who use computers as a creative partner (computer-aided composition). For computer scientists, it is a challenging application domain.

The resulting rules can be used in existing rule-based approaches to algorithmic composition where multiple rules can be freely combined, for example, in constraint-based systems (*Anders & Miranda, 2011*). Rules derived by ML and rules programmed manually can be freely combined in such systems, and rules can address various aspects (e.g. rules on rhythm, melody, harmony, voice leading, and orchestration). Potentially, ML can be used to derive rules from a given corpus of music for aspects where we do not have rules yet, for example, how to rhythmically and melodically shape the development of a phrase in a certain style.

This article describes a pilot project within the research programme described above. In this pilot, we automatically extract rules for the treatment of dissonances in Renaissance music using a corpus of movements from Palestrina masses. The treatment of such dissonances is rather well understood, which helps evaluating results. Nevertheless, this task is far from trivial, as it has to take various musical viewpoints into account (e.g. melodic interval sizes and directions, note durations, and metric positions). Results can be interesting and useful not only for musicologists and composers, but also for the commercially relevant field of music information retrieval to advance the still unsolved problem of automatic harmonic analysis of polyphonic music.

## BACKGROUND

### Inductive logic programming

Symbolic compositional rules have been extracted by ML before, specifically with inductive logic programming (ILP). ILP (*Muggleton et al., 2012*) combines logic programming with ML in order to learn first-order logic formulas from examples. Background knowledge expressed in logic programmes can be taken into account.

Inductive logic programming has been used for several musical applications. Closely related to our goal is the work of *Morales & Morales (1995)*. Their system learnt standard counterpoint rules on voice leading, namely how to avoid open parallels. Other musical applications of ILP include the learning of harmonic rules that express differences between two music corpora, specifically Beatles songs (pop music) and the Real Book (jazz) (*Anglade & Dixon, 2008*), and music performance rules for piano (*Tobudic & Widmer, 2003*) and violin (*Ramirez et al., 2010*).

Numeric relations are difficult to deduce with ILP, as logic programming in general is very restricted in expressing numeric relations. We are specifically interested in also learning numeric relations besides logic relations, because our experience with

constraint-based modelling of music composition makes us aware of their importance for compositional rules. For example, the size of melodic and harmonic intervals is important, and such quantities are best expressed numerically. Besides, we want to use learnt rules later with constraint programming, a programming paradigm with very good support for restricting numeric relations.

## Genetic programming

In this project we therefore use another approach. GP is a particular kind of evolutionary algorithm that is used for ML. In GP, a tree structure is learnt by repeated application of random changes (mutation and recombination) and by the selection of the best structures among a set of candidates (a population) according to some criterion. A candidate tree can be the representation of a computer program or a mathematical equation among other possibilities. Early seminal work on GP has been published by *Koza (1992)*, a more recent practical introduction can be found in *Poli, Langdon & McPhee (2008)*.

A particularly important application of GP is symbolic regression. Symbolic regression infers a mathematical expression that best fits the given data. The mathematical expression is unrestricted except that a specified set of building blocks is used–operators like +, or standard mathematical functions. The trees that GP builds from these building blocks are representations of such mathematical expressions. Symbolic regression is a powerful method that has been used in various areas of science and engineering (*Poli, Langdon & McPhee, 2008*), including a high-profile study where it was used to automatically deduce physical laws from experiments (*Schmidt & Lipson, 2009*).

Genetic programming has been used for music composition before. *Spector & Alpern (1994)* propose a system that automatically generates computer programmes for composing four-measure bebop jazz melodies. The generated programmes combine a number of given functions, inspired by Jazz literature, that transform short melodies from Charlie Parker in various ways. The fitness of each programme is evaluated by a set of restrictions inspired by *Baker (1988)*, which measure the balance of different aspects (e.g. tonal and rhythmic novelty).

*Johanson & Poli (1998)* also propose a system that creates computer programmes for transforming short melodies, but they allow users to interactively rate the quality of the generated music. This proved a tedious process for users. Therefore they complement the user-rating with automatic fitness raters that learn from the user ratings.

Previous applications of GP for music composition thus aimed at modelling the full composition process, where the fitness function had to judge the quality of the resulting music. Yet, the programmes resulting from the evolutionary algorithm are rather short, and they are thus limited in the compositional knowledge they can represent. Previous work therefore composed music by transforming pre-existing music.

Instead, we are interested in learning compositional rules with GP that describe only a certain aspect of the resulting music. Such rules are relevant in their own right as a representation of compositional knowledge that can be complemented with further

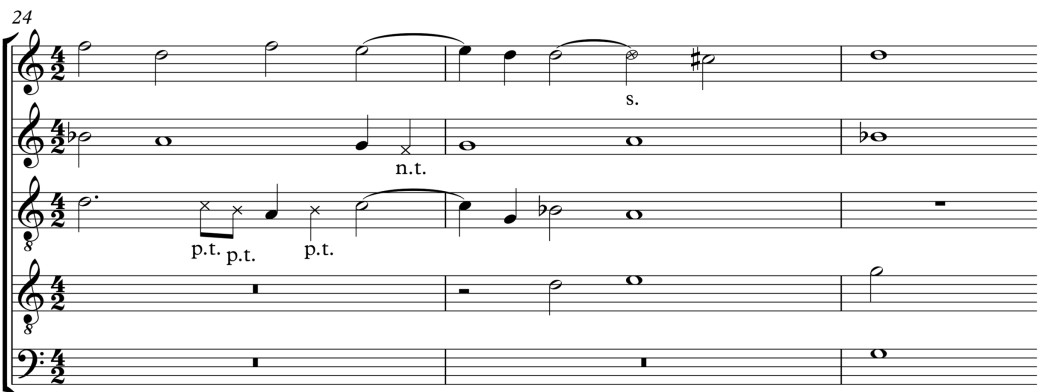

**Figure 1 Palestrina excerpt with several dissonances: several passing tones (p.t.), a neighbour tone (n.t.), and a suspension (s.), from the Agnus of missa De Beata Marie Virginis (II), measures 24–26.**

musical knowledge, for example, in music constraint programming systems with manually encoded musical rules.

In this situation, the fitness function does not need to judge musical quality. Instead, it only needs to estimate how well the resulting rule fits the given positive examples and avoids the negative examples.

As far as we know, GP has not yet been used for learning symbolic compositional rules, and therefore in this study we focus on a relatively well understood class of rules.

## Dissonances in Palestrina's music

This project studies the dissonance treatment in Palestrina counterpoint with ML by automatically generating multiple symbolic rules that each constrain the treatment of a specific dissonance category (passing tones, suspensions etc.).

*Jeppesen (1946)*, the seminal authority on Palestrina counterpoint, distinguishes three roles a dissonance can play in his music. Some dissonances are hardly noticeable on a weak beat used for connecting notes in smooth melodic lines (Jeppesen calls them dissonances as a secondary phenomenon); others occur at an accented beat and are clearly heard (dissonances as primary phenomenon); and finally—more rarely—dissonances can be used for an expressive effect.

As an example, Fig. 1 shows an excerpt from a Palestrina mass movement[1] with several dissonance categories in close proximity. All dissonances are marked with a crossed notehead, and are labelled with their dissonance category. Passing tones (p.t.) and neighbour tones (n.t.) are short notes on a weak beat that link melodic tones (secondary phenomenon). By contrast, suspensions (s.) stand out; they occur on a strong beat and typically at longer notes.

The five standard dissonance categories taught in Palestrina counterpoint classes are passing tone, neighbour tone, suspension, anticipation and cambiata (*Jeppesen, 1939*). Traditionally, these categories are introduced progressively in class for pedagogic reasons in so-called species counterpoint, for example, in the 1725 textbook by *Fux (1965)*, which is still widely used.

[1] The excerpt is from the Agnus of missa De Beata Marie Virginis (II), which is `Agnus_0.krn` in the music21 corpus, and stems from Humdrum. See also footnote 3.

An algorithm for automatically identifying dissonances in Renaissance counterpoint has been proposed by *Patrick & Strickler (1978)*, but it implements knowledge on the standard dissonance categories and therefore we instead developed a custom algorithm. The actual music of Palestrina contains further dissonance categories, as shown by computational analysis (*Sigler, Wild & Handelman, 2015*).

Counterpoint composition has often been modelled in a rule-based way, together with the relevant dissonance treatment. Already the 1957 composition *Illiac Suite*, commonly recognised as the first computer-generated composition, implemented for its second experiment and movement four-part first species counterpoint with rules similar to Renaissance counterpoint rules (*Hiller & Isaacson, 1993*). As a model for first species counterpoint, this early work sidestepped the issue of dissonance treatment, though, by simply not allowing for any dissonances. *Ebcioglu (1980)* proposed a system that implemented two-part florid counterpoint to a given cantus firmus with rules inspired by Renaissance counterpoint that stem from Joseph Marx and Charles Koechlin and are complemented by rules of the author. Standard dissonance categories like passing tones, neighbour tones and suspensions are implemented, and published results attest to the musical quality of the output. *Schottstaedt (1989)* followed the Palestrina counterpoint rules of *Fux (1965)* very faithfully and implemented all of Fux' five species for up to six voices with own rules added to get closer to Fux' actual examples.

## METHODS

For learning symbolic compositional rules we use a novel methodology that combines multiple established approaches. At first, dissonant notes are automatically labelled in a corpus of music by Palestrina with a custom algorithm. These dissonances are then automatically clustered into different dissonance categories (passing notes, suspensions etc.) with the clustering algorithm DBSCAN (*Ester et al., 1996*). Finally, a rule is learnt for each of these categories with GP. Each rule states conditions that must hold between three consecutive melodic notes if the middle note is a dissonance. The rest of this section describes each of these steps in more detail.

### Annotation of dissonances

#### A custom algorithm for dissonance detection in Renaissance music

As a first step we automatically label dissonances in the music using a custom algorithm implemented with the music analysis environment music21 (*Cuthbert & Ariza, 2010*). For our purposes, it is better for the algorithm to leave a few complex dissonance categories undetected than to wrongly mark notes as dissonances that are actually not dissonant. Note that this algorithm does not implement any knowledge of the dissonance categories known to occur in Palestrina's music.

The analysis first 'chordifies' the score, that is, it creates a homorhythmic chord progression where a new chord starts whenever one or more notes start in the score, and each chord contains all the score notes sounding at that time. The result of the 'chordification' process on its own would loose all voice leading information (e.g. where voice crossing happens), but our algorithm processes those chords and the original

polyphonic score in parallel and therefore such information is retained. The algorithm tracks which score notes form which chord by accessing the notes with the same start time (music21 parameter *offset*) as a given chord.

The algorithm then loops through the chords. If a dissonant chord is found,[2] it tries to find which note(s) make it dissonant by testing whether the chord becomes consonant if the pitches of these note(s) are removed.

Dissonances are more likely to occur on short notes in Palestrina, and sometimes multiple dissonant tones occur simultaneously. The algorithm tests whether individual notes (and pairs of simultaneously moving notes) are dissonant. The algorithm progresses in an order that depends on the notes' duration and a parameter *max_pair_dur*, which specifies the maximum duration of note pairs tested (in our analysis *max_pair_dur* equalled to a half note). In order to minimise mislabelling dissonances, the algorithm first tests all individual notes with a duration up to *max_pair_dur* in increasing order of their duration; then all note pairs in increasing order of their duration; and finally remaining individual notes in order of increasing duration.

Suspensions are treated with special care. Remember that the algorithm processes the 'chordified' score and the actual score in parallel. As a preprocessing step, the algorithm merges tied score notes into single notes to simplify their handling. When the algorithm then loops through the chords and finds a dissonant note that started before the currently tested chord, it splits that note into two tied notes, and only the second note starting with the current chord is marked as dissonant.

In order to avoid marking notes wrongly as dissonances, the algorithm does not test the following cases: any note longer than *max_diss_dur*, a given maximum dissonance duration (we set it to a whole note); and any suspension where the dissonant part of the note would exceed the preceding consonant part, or it would exceed *max_diss_dur*.

For this pilot we arbitrarily selected the first 36 Agnus mass movements from the full corpus of Palestrina music that ships with music21.[3] All examples in that subcorpus happen to be in $\frac{4}{2}$ 'metre' (tempus imperfectum, denoted with cut half circle), but our method does not depend on that.

### Evaluation of the dissonance detection algorithm

We evaluated results by manually 'eyeballing' a sample of scores with marked dissonances. The dissonance detection appears to work rather well; only very few notes were found wrongly labelled as a dissonance during our manual examination.[4]

Figure 1 shows an example where dissonances would not be correctly labelled by our algorithm. The first two passing tones (eighth notes) are correctly labelled in Fig. 1, but our algorithm would instead label the D in the soprano as a dissonance. The problem here is that when the two correct dissonances are removed, the resulting 'chord' A–D is a fourth, which is still considered a dissonance in Renaissance music. Instead, if the soprano D is removed, the remaining chord C–A happens to be consonant. Also, sometimes a suspension is not correctly recognised and instead a wrong note is labelled, where the voice proceeds by a skip in shorter notes.

[2] A dissonant chord is any chord for which music21's `Chord.isConsonant()` returns false. That function checks for two pitches whether the interval is a major or minor third, sixth or perfect fifth, and for three pitches whether the chord is a major or minor triad that is not in second inversion.

[3] Casimiri edition (*Da Palestrina, 1939*), encoded by John Miller and converted to Humdrum by Bret Aarden.

[4] Unfortunately, we have no way to automatically estimate the quality of our dissonance detection. Nevertheless, the interested reader can confirm in music notation the quality of the dissonance detection by examining the results in the data for this paper (available under digital object identifier (DOI) 10.5281/zenodo.2653502) in the folder Preprocessing Results, which includes MusicXML files of all pieces we used, and where detected dissonances are marked by x-shaped note heads.

**Table 1 The features of the automatically derived clusters match traditional dissonance categories.**

| Cluster | Dissonance category | 1st interval | 2nd interval | Metric position | Duration |
|---------|---------------------|--------------|--------------|-----------------|----------|
| C0 | Passing tones down | Step down | Step down | Weak beat | Up to half note |
| C1 | Passing tones up | Step up | Step up | Weak beat | Up to half note |
| C2 | Suspension on beat 3 | Repetition | Step down | Strong beat 3 | Quarter or half note |
| C3 | Suspension on beat 1 | Repetition | Step down | Very strong beat 1 | Quarter or half note |
| C4 | Lower neighbour tone | Step down | Step up | Weak beat | Up to half note |

To be on the safe side for our purposes of learning dissonance treatment, the algorithm therefore does not label all dissonances. For example, occasionally more than two dissonances occur simultaneously in Palestrina, for example, when three parts move with the same short rhythmic values or in a counterpoint of more than four voices. If the algorithm does not find tones that, when removed, leave a consonant chord, then no note is labelled as a dissonance (though the chord is marked for proofreading). Excluding dissonances with the parameter *max_diss_dur* avoided a considerable number of complex cases and otherwise wrongly labelled notes.[5]

Due to these precautions (avoiding wrongly labelled dissonances), due to the corpus (selected Agnus movements) and the frequency of certain dissonance categories in that corpus, only a subset of the standard dissonance categories were detected and learnt as rules in this project (see Table 1). This point is further discussed below in the section on the evaluation of the clustering. Also, because the wrongly labelled dissonances are relatively rare and do not show regular patterns, the cluster analysis sorts these cases into an outlier category, which we did not use for rule learning, further improving the quality of the marked dissonance collections.

### Data given to machine learning

Instead of handing the ML algorithm only basic score information (e.g. note pitches and rhythmic values), we provide it with background knowledge (like melodic intervals and accent weights as described below), and that way guide the search process. For flexibility, we considered letting the ML algorithm directly access the music21 score data with a set of given interface functions (methods), but extracting relevant score information in advance is more efficient.

Once dissonances are annotated in the score, only melodic data is needed for the clustering and later the ML. For simplicity we only used dissonances surrounded by two consonant notes (i.e. no consecutive dissonances like cambiatas).

In order to control the differences in 'key' between pieces in the corpus, our algorithm computes 'normalised pitch classes', where 0 is the tonic of the piece, 1 a semitone above the tonic and so forth. For this purpose we need to compute the 'key' of each piece. We are not aware of any algorithm designed for classifying Renaissance modes, and we therefore settled on instead using the Krumhansl-Schmuckler key determination algorithm (*Krumhansl, 1990*) with simple key correlation weightings by *Sapp (2011)*.

[5] Interested readers can investigate the unlabelled dissonances in music notation. In the MusicXML files mentioned in the previous footnote, the result of the 'chordification' process is included as an additional stave and all dissonances are always labelled in that stave. If the staves of the actual voices do not contain any simultaneous note with x-shaped note head, then that dissonance is not labelled in the score.

[6] As future work, it might be worth exploring whether the low-complexity weights that Sapp proposed for major and minor—basically assigning 0 to all non-scale degrees and 1 to all scale degrees, but 2 to the tonic and fifth—could be adapted for Renaissance modes by assigning 2 to their respective tonics (and the fifths as appropriate).

While this approach is not able to recognise modes, it is at least able to control the differences in accidentals between pieces.[6]

We determine accent weights using music21's function *getAccentWeight*, where the strongest beat on the first beat of a measure has the weight 1.0; strong beats within a measure (e.g. the third beat in $\frac{4}{2}$ 'metre') the weight 0.5; the second and fourth beat in $\frac{4}{2}$ metre the weight 0.25 and so on (*Ariza & Cuthbert, 2010*).

Intervals are measured in semitones, and durations in quarter note lengths of music21, where 1 means a quarter note, 2 a half note and so on.

For simplicity we left ties out in this pilot. Suspensions are simply repeated tones, they are not tied over.

## Cluster analysis of dissonance categories

### Analysis With DBSCAN algorithm

For each dissonance, we provide the clustering algorithm with the following features: the sum of the durations of the previous, current, and next note (the reason for including this feature instead of including all note durations is explained in the following 'Discussion' section); the melodic interval from the previous note (measured in semitones), and the interval to the next note; the 'normalised' pitch class of the dissonance (i.e. 0 is the 'tonic' etc.); and the accent weight of the dissonance. Before clustering, all data for each feature is normalised such that its mean is 0.0 and its standard deviation is 1.0.

The data is clustered using the DBSCAN algorithm (*Ester et al., 1996*) as implemented in the scikit-learn library (*Pedregosa et al., 2011*). This clustering algorithm does not require setting the number of clusters in advance, and can find clusters of arbitrary shape as it is based on the density of points. Here, we set the minimum number of neighbours required to identify a point as a core cluster point to 10, and the maximum neighbour distance to 0.7 based on initial runs and the desire to keep the number of clusters in a reasonable range. Points that lie in low density regions of the sample space are recognised as outliers by DBSCAN, and are ignored in the subsequent rule learning step.

### Clustering results and discussion

In order to evaluate the clustering results, we automatically labelled each dissonance in the score with its dissonance category (cluster number), and then created a new score for each cluster number into which all short melodic score snippets that contain this dissonance were collected (one-measure snippets, except where the dissonance occurs at measure boundaries). We then evaluated the clustering results by eyeballing those collections of score snippets[7].

[7] The clustering results, along with all algorithms created, can also be found in the Supplemental Data, DOI: 10.5281/zenodo.2653502.

Initially, the importance of note durations for the clustering was rated too highly, because the clustering took more duration parameters into account (one for every note) than other parameters (e.g. pitch intervals between notes). As a result, one cluster contained primarily dissonances at half notes and another dissonances at shorter notes, which was not useful for our purposes. Therefore, we aggregated the duration information, and adjusted the DBSCAN parameters as described above, after which clustering worked very well.

In the selected corpus only the following main dissonance categories are found: passing tones downwards on a weak beat (863 cases); passing tones upwards on a weak beat (643 cases); suspensions on the strong beat 3 in $\frac{4}{2}$ 'metre' (313 cases); suspensions on the strong beat 1 (265 cases); and lower neighbour tones on a weak beat (230 cases).

Table 1 summarises the distinguishing features of the dissonance categories as found in clusters C0–C4, for which we learnt rules. Each row in the table describes a separate dissonance category as recognised by the clustering algorithm. The first interval indicates the melodic interval into the dissonance, and the second the interval from the dissonance to the next note. Metric position and duration are features of the dissonant note itself.

Other dissonance categories like upper neighbour tones, anticipations and cambiatas do not occur in the ML training set. Either they do not exist in the first 36 Agnus mass movements of the music21 Palestrina corpus that we used, or they were excluded in some way. We only use dissonances surrounded by consonances (which excludes cambiatas). Also, we did not use the set of outliers (189 cases), which as expected, has no easily discernible common pattern. Among these outliers are wrongly labelled dissonances, upper neighbour tones, and a few further cases of the above main categories. There are also two small further clusters with lower neighbour tones (C5, 25 cases), and passing tones upwards (C6, 11 cases) that were discarded as they were considered to be too small, and cover categories that are already covered by larger clusters.

## Learning of rules

### Training set

To initiate rule learning, our algorithm compiles for each identified cluster (dissonance category) a set of three-note-long learning examples with a dissonance as middle note. All dissonances that have been assigned to that particular cluster are used as positive examples.

Then, four sets of negative examples are generated. Note that the generation of negative examples does not take any knowledge about the problem domain into account. A similar approach can also be used for learning rules on other musical aspects. The first set is a random sample of dissonances that have been assigned to other clusters. The second set is a random sample of three-tone-examples without any dissonance taken from the corpus. The third set consists of examples where all features are set to random values drawn from a uniform distribution over the range of possible values for each feature. The fourth set consists of slightly modified random samples from the set of positive examples. Two variations are generated for each chosen example. Firstly, either the interval between the dissonant tone and the previous note or the interval to the next note is changed by ±2 semitones (with equal probabilities). Secondly, one of the three note durations is either halved or doubled (with equal probabilities). Both modifications are stored separately in the fourth set of negative examples.

The algorithm aims to create 20% of the number of positive examples for each set of negative examples, but will generate at least 200 examples (100 for the first set due to possible low availability of these examples) and at most 500. These numbers represent

mostly a trade-off between accuracy of training/measurement and computation time, and we expect a similar performance if these values are changed within reasonable ranges.

Once all example sets have been constructed, each example receives a weight such that the total weight of the positive examples is 1.0, and the total weight of each of the four sets of negative examples is 0.25 (within each set, the weights are the same for all examples). When measuring classification accuracy of a rule during the learning process, each positive example that is classified correctly counts +1 times the example weight, whereas each negative example that is erroneously classified as positive example counts −1 times the example weight. Thus, the accuracy score is a number between −1.0 and 1.0, with 0.0 expected for a random classifier.

Please note that with a low probability a randomly generated negative example can be the same as a positive example. Here, we consider this as a small amount of noise in the measurement, but for future experiments it is possible to filter these out at the expense of run time.

### Learning process

We use strongly typed GP as implemented in the Python library DEAP (https://github.com/deap/deap) (*Fortin et al., 2012*) with the types float and Boolean (integers are translated to floats). The following functions can occur as parent nodes in the tree that represents a learnt rule.

Logical operators: ∨ (or), ∧ (and), ¬ (not), → (implication), ↔ (equivalence)
Arithmetic operators and relations: +, −, · (multiplication), / (division), − (unary minus), =, <, >
Conditional: *if_then_else*(⟨*boolean*⟩, ⟨*float*⟩, ⟨*float*⟩)

Terminal nodes in a 'rule tree' can be either input variables (like the duration of a note or the interval to its predecessor or successor) or ephemeral random constants (constants whose values can be changed by mutations). The following input variables can be used in the learnt rules: the duration of the dissonant note ($duration_i$), its predecessor ($duration_{i-1}$) and successor ($duration_{i+1}$); the normalised pitch class of the dissonance; the intervals[8] between the dissonance and its predecessor ($interval_{pre}$) and successor ($interval_{succ}$); and the accent weight of the dissonance ($accentWeight_i$). As for the ephemeral random constants, there are the Boolean constants *true* and *false*, as well as integer constants in the form of values between 0 and 13. The notation given here is the notation shown later in learnt rule examples.

There are many choices of operators and parameters that can be used with GP. Here, we follow standard approaches that are commonly used in the GP practitioners' community, and/or are DEAP defaults, unless otherwise noted. The population is created using ramped half-and-half initialisation, after which at each generation the following operators are applied. For selection, we use tournament selection with a tournament size of 3. For mutation, there is a choice between three operators: standard random tree mutation (95% probability), a duplication operator that creates two copies of the tree and connects them using the ∧ operator (2.5%), and a similar duplication operator using the

[8] A melodic interval is always computed as the interval between a given note and its predecessor and positive when the next note is higher.

∨ operator (2.5%). For recombination, there is again a choice between standard tree exchange crossover (95%), an operator that returns the first individual unchanged, and a combination of the first and second individual using the ∧ operator (2.5%), and a similar operator using ∨ (2.5%). While random tree mutation and tree exchange crossover are commonly used, we designed the other operators to encourage the emergence of rules that are conjunctions or disjunctions of more simple rules, which is a useful format for formal compositional rules. Without these operators, it would be extremely unlikely that a new simple rule could be evolved without disrupting the already evolved one, or that different already evolved rules could be combined as a whole. A static depth limit of 25 is imposed on the evolving trees to avoid stack overflows and exceedingly long execution times.

A population of 100 individuals is evolved for 1,000 generations. The fitness assigned to an individual is 10 times its accuracy score (described above) minus 0.001 times the size of its tree. That way, among two rules with equal classification accuracy, the more compact rule has a slight fitness advantage. We introduced this as a measure to slow down the growth of the trees during evolution (known as 'bloat control' in the field of GP, although the effect of this particular measure is not very strong). We performed five runs for every cluster. They use the same set of learning examples, but find different rules nonetheless due to the stochastic nature of GP.

After a run is finished, the best rule evolved in that run is output together with its classification accuracy scores[9].

[9] The evolved rules from all runs, together with their qualitative evaluatiuon, can be found in the file 'Resulting rules.pdf' in the Supplemental Data, DOI: 10.5281/zenodo.2653502.

# RESULTS

## Quantitative evaluation

The quality of the rule learning process as implemented by GP is evaluated by measuring the accuracies of the best evolved rules (see Fig. 2). It can be seen that the accuracies for positive examples are better than 98% in most cases, the accuracies on negative examples from other clusters are mostly better than 99%, the accuracies on negative examples without dissonances are mostly better than 94%, the accuracies on random counterexamples are close to 100%, and the accuracies for modified positive examples are mostly better than 94% but around 89% for the first cluster. When plotting overall accuracy scores against the sizes of the rules' corresponding GP trees (Fig. 3), it can be seen that rules for the same cluster achieve similar accuracy scores despite different sizes. However, across clusters, there seems to be a negative correlation between accuracy and rule size. The most plausible explanation seems to be that larger clusters are more difficult to describe, resulting both in larger rule sizes and lower accuracy scores (Fig. 4). The accuracy of some negative examples (i.e. examples without dissonances, and modified positive examples) might be lower, because some of them may accidentally match the pattern of positive examples.

## Qualitative evaluation

We evaluated the suitability of the evolved rules for describing dissonances by using them as melodic constraints in small constraint problems implemented with the music constraint system Cluster Engine (http://github.com/tanders/clusterengine), which is a

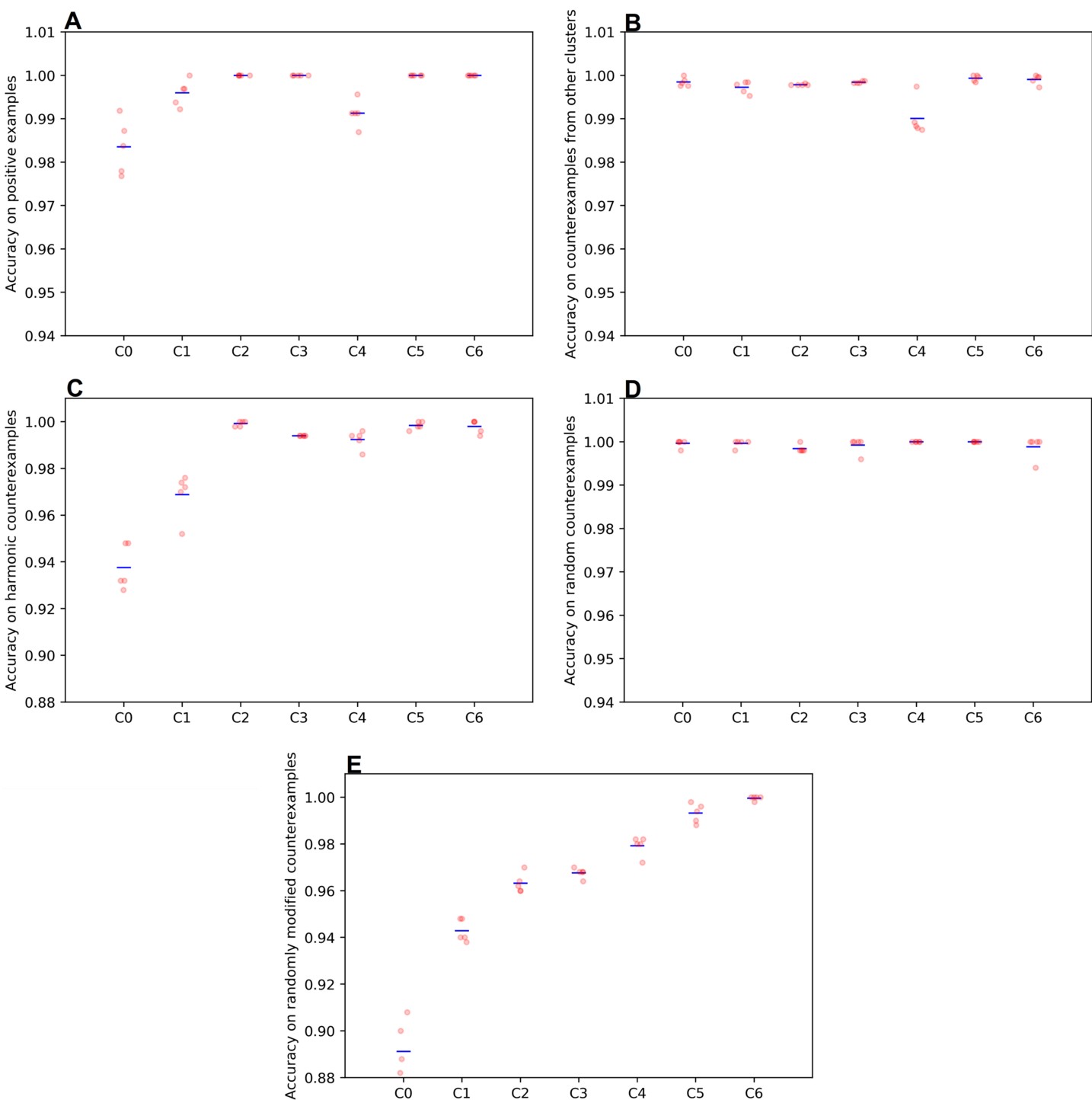

**Figure 2 Accuracies achieved by the champions of the genetic programming runs on the various parts of the training set.** (A) postive examples; (B) counterexamples from other clusters; (C) harmonic counterexamples; (D) random counterexamples; (E) randomly modified positive examples. The means are indicated by blue bars. C0–C6 denotes the clusters found by DBSCAN. These clusters correspond to different dissonance categories (see Table 1). C5 and C6 are two small clusters that were disregarded in the end (see discussion at the end of section 'Clustering Results and Discussion' above).

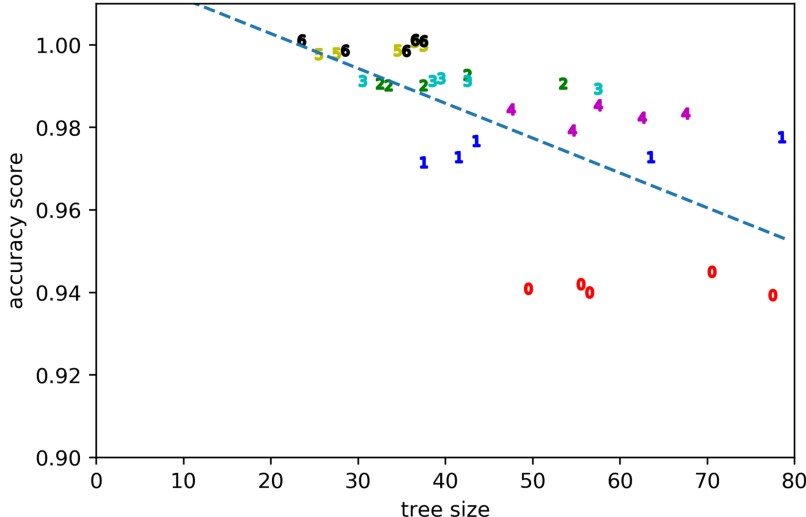

**Figure 3 Accuracy score vs tree size for the evolved solutions from all runs.** Clusters are denoted by their respective numbers.

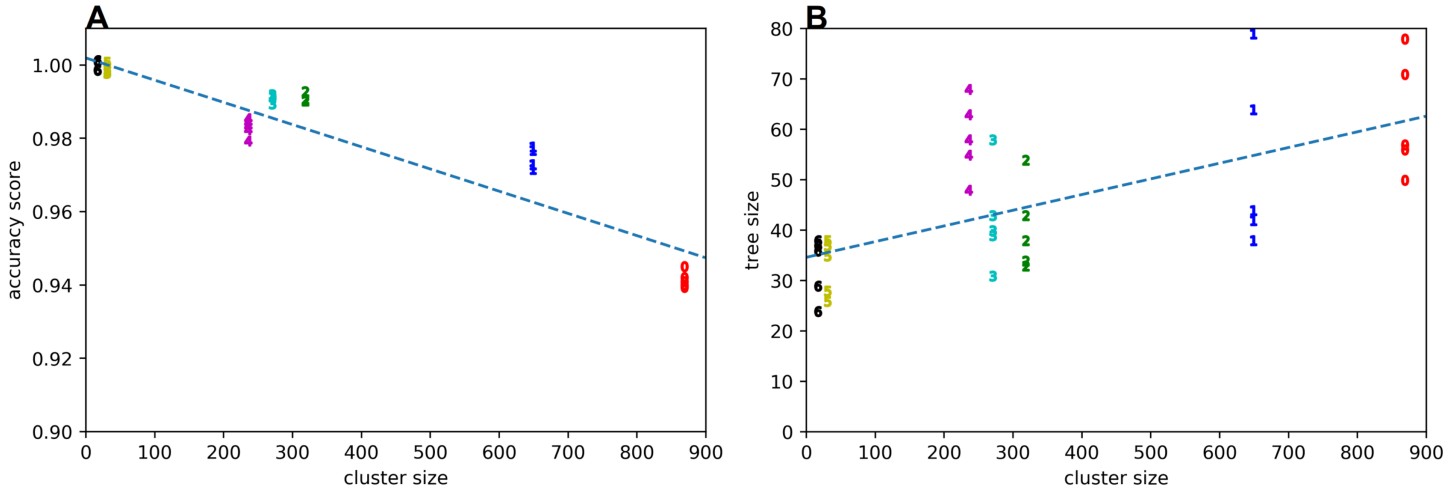

**Figure 4 Accuracy score (A) and tree size (B) vs cluster size.** Clusters are denoted by their respective numbers.

revision of the solver PWMC (*Sandred, 2010*). Both these solvers are libraries of the visual programming and composition environment PWGL (*Laurson, Kuuskankare & Norilo, 2009*). The constraint problems consist of only three notes with the middle note as the dissonance and surrounding rests as padding so that these notes can occur freely on any beat.

For each learnt rule (five per cluster resulting from the five runs reported above) we generated 15 random solutions (an arbitrary number). We examined these solutions in common music notation, and appraised how well they complied with the respective dissonance category. Specifically, we checked whether the metric position and duration of the middle note (the dissonance) and the melodic intervals into and from this note are appropriate.

**Table 2 Qualitative evaluation summary: greatest deviation found between features of positive examples (see Table 1) and solutions to best rule for each cluster.**

| Cluster | Category | 1st interval | 2nd interval | Metric position | Duration |
|---------|----------|--------------|--------------|-----------------|----------|
| C0 | Passing tones down | None | None | None | None |
| C1 | Passing tones up | None | None | None | None |
| C2 | Suspension on beat 3 | None | None | None | Small[a] |
| C3 | Suspension on beat 1 | Small[b] | None | None | Small[a] |
| C4 | Lower neighbour tone | Small[c] | None | None | None |

**Notes:**
[a] Rule also allows for eighth note (ideally it would only allow for a quarter or half note).
[b] Rule also allows for a minor second down (ideally it would only allow for pitch repetition).
[c] Rule also allows for a repetition (ideally it would only allow for a downward step).

$$duration_{i-1} < duration_i + 3 \tag{1}$$
$$\wedge\ accentWeight_i < 0.5 \tag{2}$$
$$\wedge\ 6 \cdot accentWeight_i < duration_{i-1} \tag{3}$$
$$\wedge\ accentWeight_i < interval_{pre} \tag{4}$$
$$\wedge\ accentWeight_i < interval_{succ} \tag{5}$$
$$\wedge\ duration_i < 3 \tag{6}$$
$$\wedge\ accentWeight_i \cdot duration_{i+1} < duration_i \tag{7}$$
$$\wedge\ interval_{pre} < 3 \tag{8}$$
$$\wedge\ interval_{succ} < 3 \tag{9}$$

**Figure 5 Learnt rule example from C1: passing tones upwards.**

For each cluster (dissonance category) at least one learnt rule constrains the treatment of the dissonant middle note in a way that either fully complies with the features of the corresponding positive examples (see Table 1 again), or is at least very close. In other words, this 'best rule' per cluster works either perfectly or at least rather well when used as a constraint for its dissonance category. Table 2 shows how accurate the best rule of each dissonance category is by reporting the greatest deviation found in any solution among a set of 15 random solutions. The full qualitative evaluation results—together with the raw rule results—are available with the data provided with this paper (under *Resulting Rules and Evaluation*).

## Examples of learnt rules

To give a better idea of the kind of rules learnt, Figs. 5 and 6 show two examples. The rule of Fig. 5 constrains passing tones upwards and that of Fig. 6 suspensions on beat 1. These specific rules have been selected, because they are relatively short. Both rules are the best out of their set of 5 in the sense just discussed above, and the qualitative evaluation of their results is included in Table 2. The rules generated by DEAP were slightly simplified manually and with Mathematica, and translated into standard mathematical notation for clarity.

$$2 \geq |interval_{succ}|$$
$$\wedge \ accentWeight_i \geq 1$$
$$\wedge \ (2 < duration_i \vee duration_{i-1} \geq 2)$$
$$\wedge \ (2 \geq duration_i \vee duration_{i-1} < 2)$$
$$\wedge \ interval_{pre} < accentWeight_i$$
$$\wedge \ interval_{pre} > interval_{succ}$$

**Figure 6 Learnt rule example from C3: suspension on beat 1.**

As an example, let us analyse the first rule, which constraints upwards passing tones (Fig. 5). The other rule in Fig. 6 can be analysed similarly. Remember that for this dissonance category the dissonance occurs on a weak beat, both intervals lead upwards stepwise, and its duration is up to a half note (Table 1). This rule constrains all those aspects exactly (Table 2). The names of the input variables and their possible values have been explained above,[10] but we will briefly revise them below for the reader's convenience. Note that the learnt rules contain some bloat—expressions that are irrelevant or redundant. In our analysis we will focus on the meaningful expressions of the rule.

Remember that each rule states conditions that must hold between three consecutive melodic notes if the middle note is a dissonance. The dissonance occurs on the note with the index $i$, for example, $accentWeight_i$ is the accent weight of the dissonance. The rule in Fig. 5 constrains dissonances to a weak beat. For the first beat of a measure, the accent weight is 1.0, for the third beat in $\frac{4}{2}$ it is 0.5, of the second and forth beat it is 0.25 and so on. The rule constrains $accentWeight_i$—the weight of the dissonance—to less than 0.5, that is, to a weak beat, see line (2) of the rule.

The rule enforces that both the melodic interval into the dissonance and out of it, $interval_{pre}$ and $interval_{succ}$ (measured in semitones) are positive (upwards). The rule happens to express this by constraining that these intervals are greater than $accentWeight_i$, see Eqs. (4) and (5), and $accentWeight_i$ is always greater than 0 by its definition. Both intervals are less than 3, see Eqs. (8) and (9). So, in summary the intervals are positive (upwards), but at most two semitones (steps).

The duration must be a half note or shorter. Durations are measured in music21's quarterlengths, where 1 represents a quarter note. The duration of the dissonance must be less than 3, which corresponds to a dotted half note (6), hence it can be a half note at most.

## DISCUSSION

During the course of our project we gradually improved the resulting rules. The negative examples in the training set for the rule learning have a great impact on the accuracy of resulting rules. For example, the inclusion of slightly modified transformations of positive examples clearly improved the accuracy as compared to preliminary experiments. A closer investigation into the impact of automatically generated negative examples on the accuracy of resulting rules could lead to further improvement. For example, so far we only used slight random variations of the note durations and melodic intervals to

[10] The variable names where introduced above when discussing terminal nodes in the subsection 'Learning process' and their possible values earlier in the subsection 'Data given to machine learning'.

generate negative examples, but variations of further parameters could also be useful (e.g. negative examples with shifted metric positions could also restrict syncopations).

Multiple rules learnt for the same cluster differ in their accuracy when used as a constraint for music generation: the music generated with these rules (during qualitative evaluation) can be more or less close to the features of the positive examples (see Table 1). Table 2 only reports the accuracy of the best rule per cluster. Some other rules for the same cluster are much less accurate, but nevertheless obtain a very similar overall weighted score in the learning process. Currently, we lack an algorithm for measuring the accuracy of a rule in terms of how similar generated music restricted by that rule is compared with its positive examples. Such an algorithm would be very useful to contribute to the fitness calculation of rules during the learning process.

The accuracy of resulting rules can also be improved by taking further background knowledge into account. For example, some resulting rules allow for syncopations in dissonance categories where these would not occur in Palestrina, for example, at a passing tone. Providing the rule learning algorithm with an extra Boolean feature whether the dissonant note is syncopated or not will likely avoid that.

A further improvement could probably be obtained by post-processing large clusters generated by DBSCAN with another clustering algorithm that is not based on density alone, or by DBSCAN with a smaller setting for maximum neighbour distance, to split them up into smaller clusters, for which learning a suitable rule should be easier.

From the perspective of GP research, we demonstrate a simple yet powerful way of performing symbolic regression on heterogeneous data. A combination of GP and clustering is used to achieve that. These two techniques have occasionally been used together for various purposes. Some similarity to our approach can be found in the work by *Kattan, Agapitos & Poli (2010)*, who use a top-level genetic programmming system to project data from a complex problem domain onto a two-dimensional space, then apply $k$-means clustering several times with different target cluster numbers to subdivide the problem set, then use the best clustering (as determined by a clustering quality measure) to run a lower level GP system on each cluster. This method was introduced against a background of techniques other than clustering that had already been applied to what is termed problem decomposition for GP. The authors demonstrate superior performance on a set of benchmarks problems against GP without clustering. Later, *Kattan, Fatima & Arif (2015)* extend this approach for event prediction in time series. By using a density based clustering algorithm, and applying it directly on the input data, we achieve something similar with a much simpler system. This is partly due to the fact that the density parameter is not difficult to set. Of course, if it were more difficult to set, clustering quality metrics could also be used for that purpose along with the manual evaluation of clustering quality based on domain knowledge that we have applied here.

## CONCLUSIONS

Human composition education and composition practice commonly combine guidance from compositional rules and insights learnt from musical examples. By contrast, while rule-based approaches on the one hand, and approaches based on ML on the other hand

have been often employed in the field of algorithmic composition, these approaches have rarely been combined.

In this paper, we propose a method that automatically extracts symbolic compositional rules from a music corpus. This method is based on GP, a ML technique using an evolutionary algorithm. The resulting rules can be analysed by humans or used in rule-based algorithmic composition systems. In this study, we extracted rules that detail the dissonance treatment in compositions by Palestrina. We derived rules for the following five dissonance categories (automatically derived clusters): passing tones on a weak beat upwards and downwards; lower neighbour tones on a weak beat; and suspensions on the strong beat one and beat three in $\frac{4}{2}$ 'metre'.

Learnt rules typically match between 98% and 99% of the positive training examples, while excluding between 89% and 100% of the counterexamples depending on counterexample category and cluster (dissonance category), with better results for smaller clusters. Learnt rules describe melodic features of the dissonance categories very well, though some of the resulting best rules allow for minor deviations compared with the positive examples (e.g. allowing the dissonance category of suspension to occur on shorter notes as well).

## ACKNOWLEDGEMENTS

We want to thank Dmitri Tymoczko, who suggested the initial idea for an algorithm that detects dissonances in Renaissance music. We are also grateful for the detailed and valuable feedback of two reviewers.

### Funding

The authors received no funding for this work.

### Competing Interests

The authors declare that they have no competing interests.

### Author Contributions

- Torsten Anders conceived and designed the experiments, performed the experiments, analysed the data, prepared figures and/or tables, performed the computation work, authored or reviewed drafts of the paper, approved the final draft.
- Benjamin Inden conceived and designed the experiments, performed the experiments, analysed the data, prepared figures and/or tables, performed the computation work, authored or reviewed drafts of the paper, approved the final draft.

### Data Availability

Torsten Anders, & Benjamin Inden. (2019, October 17). Supplemental files for article: Machine learning of symbolic compositional rules with genetic programming: Dissonance treatment in Palestrina. Zenodo. DOI 10.5281/zenodo.3538295.

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
