# Peer review of "Machine learning of symbolic compositional rules with genetic programming: dissonance treatment in Palestrina"

_PeerJ Computer Science, doi:10.7717/peerj-cs.244_

## Round 0.1 · original submission · Minor Revisions

Please follow the recommendations of both reviewers, and prepare a response letter explaining how you addressed them.

Reviewer 1 ·

Basic reporting

On English:
Some typos in the abstract: “we us all other dissonances” -> “we use all other dissonances” and some shorthand also that could be made clearer. Abstract ll 14-16 could be broken into multiple sentences; and “with each other and *with* manually programmed rules” 16-17: “We chose genetic programming for our machine learning technique” I’m not sure what “as positive examples” means in l. 23 – “as examples used to train the algorithm in a particular dissonance category”? This seems redundant. The next lines 24-25 seem more important: “We also trained the algorithm in many types of negative examples such as…” l. 27 “category *of* suspension”

The cumbersome reading of the abstract almost made me not agree to review the paper; I hope others don’t decide not to read it for the same reason.

The introduction’s writing and later is stronger and better edited. But there are still errors that make reading hard – careful rereading by a native speaker is crucial:
35: methods OF algorithmic composition
144-5: “the algorithm better leaves” -- “an algorithm that leaves a few complex dissonance categories undetected is stronger than one that wrongly marks notes…” or something like that?
181: “To be at the save side” – “To be on the safe sade”

On Literature references, sufficient field background/context:
The relevant literature is addressed and cited well. However, the original encoders of the Palestrina corpus should be mentioned as well as the editions from which they are encoded. (There are errors in encoding and in parsing that are known to people involved in the original project and the music21 project; I doubt they are sufficient to affect the end results however)

On Professional article structure, figures. Raw data shared.:
Raw data is shared -- in abundance. Version of software used should be noted. Some issues w/ figures (discussed below) but in general very professional.

Figure 1 includes in addition to “p.t.” “n.t.” and “s.” also “p.t.s”. This could be read as “p.t.” followed by “s.” but the presumably correct reading as multiple passing tones is not included

On Clear defs of all terms, theorems and proofs:
Very clear in general.

A pass, but some revision needed

Experimental design

The research question is well defined, and how this work on genetic programming will advance the field of computational music research is clearly stated and well argued (well done!).

There are no ethical concerns raised by the paper.

38–42 are key and important parts of the paper (and might be highlighted in the abstract): rules that can be studied by humans are important for acceptance of this work in the discipline of music theory (as an interdisciplinary paper should aim for). Cuthbert/Ariza/Friedland 2011 p. 391 (http://ismir2011.ismir.net/papers/PS3-6.pdf), among others, discuss the importance of understanding how a decision was made in the context of classification of musical features. Very important and good point.

148 It is unclear what the custom algorithm (after chordification) does in cases where the voice leading of the chordified version is unclear.

153-159: this seems a VERY smart way of working. Applause.

The description of handling of suspensions (160-162) is still unclear to me though. After chordification, the dissonant pitches would already be tied to a previous pitch.

163-166: There are some cases in the Palestrina corpus of incorrect clefs (namely treble8vb as treble, causing fifths to become fourths). The authors are cautioned to check their corpus manually against the original encodings; all of which are on IMSLP. The authors should also be sure to be on the latest release of music21, as prior versions did not necessarily parse properly the (partially incorrectly encoded) krn files where the number of voices in Agnus II is fewer than in Agnus I, but the original (usually five-voice) structure is restored in Agnus III. This can make for a large number of dissonances.

211: were melodic intervals given in semitones, as diatonic steps, or (perhaps best) in Hewlett’s base-40 interval encoding? This information is needed to replicate. (I’m guessing from Table 1 later that it’s diatonic steps)

253: are random values that match the positive examples excluded from set 3? Or are they too few to care about? (ah, this is mentioned in 282-83. I think they should be removed).

267: again the interval should be described as diatonic or semitonal?

285–325: my knowledge of this part of the paper is low. It looks good to me, but please weigh other reviewers’ comments more highly for this section (Genetic Programming)

317-18: “The fitness assigned … is 10 times its accuracy…plus 0.001 times the size of the tree” – should this be minus 0.001 times the size of the tree? It seems to reward bloated trees?

Validity of the findings

132 – unclear why the further dissonance categories are irrelevant; they certainly do occur in the Palestrina Mass corpus, and at this point in the paper it is hard for the reader to know why rejection by the algorithms makes evidence irrelevant. (ah, l. 168 gives the answer to what the corpus is; that such dissonances do not occur is possible, but not certain)

171-175: not enough information is given here to verify the authors’ claim that the detection works rather well. Can this data be provided (or see next comment)

183–186: a reader would be interested in a list of these occurrences of dissonances that cannot be automatically classified by the custom algorithm. They would be of interest for music theorists and Renaissance scholars in addition to future researchers who may want to improve the algorithm.

196–199 is the most problematic passage I have yet read: the “key” detection of a Palestrina mass movement is likely to be very faulty as the Krumhansl-Schmuckler key determination algorithm (which should get citation) and the Sapp weights were designed for tonal music of the common practice period (1650–1880) and not for this period. I believe I have seen, in other forums, the authors’ search for an algorithm that would classify Renaissance mode better, and I believe they can say that no such algorithm yet exists, and they should be able to satisfy me that the KS algorithm is the best thing available. But the argument needs to be made.

215 – what does it mean to normalize the pitch class of the dissonance? does rotating pitch classes so, say G = 0, affect the results?

221 – the total number of points in each cluster are noted below, but the total number of points recognized as outliers should be noted. (it’s at 246)

228-233 is very well explained.

Figure 2 is hard to read – the blue bars could be made more prominent and the dots bigger as well. A note that each pink dot represents one run could be added.

The bigger issue though is that C5 and C6 are included in the diagrams but have not been explained in the text yet. What are they? [Ah! From the scripts I discover they are the cases in l. 249; this should be explained]

Figure 3 – why not plot #s 0 0 0 0, 1 1 1 1, etc. instead of abstract figures that I need to return to the caption often to read? The switch from 0-1 to 0-10 score is also confusing.

Figures 5 and 6 should explain again what cluster number they refer to – we’re going back and forth between cluster numbers and verbal descriptions.

Returning to 293–300 in light of Figure 5–6: 295: is the “dissonant” note necessarily dissonant? l. 263 seems to imply it could be simply a middle note. Figure 5 line 1: I read the right side as duration(i+3) and was wondering why the duration of the note three notes later was being taken into account. More space or something to show that it is duration(i) + 3 would help. It took me some time to note that “dissonance status” was NOT an input variable, and that we are only looking at linear (melodic) segments. If this is true, then the results are even stronger, but either way, this point should be made very clear in the article.

355 – it is unclear here and in table 2, what “can be” means – does this mean that “the generated rule/solution allows the note to be an eighth note” or “the received rule allows the note to be an eighth note but the generated solution does not”? The direction of the deviation is important.

The Qualitative evaluation (339-356) is quite cursory and there is not enough information given here to evaluate the accuracy of the results, I’m afraid.

374 – I’m not sure what an “easy beat” is. Is this a mistranslation for “simple beat” (which would also need explanation), or a integer number beat? It is not an English language term in UK or US.

381: “We leave that to the reader” – the reader does not know the grouping – does the ^ at the beginning of the line constrain the entire line or not? So are lines 2-3 to be read as
“accentWeight >= 1 AND ( 2 < duration(i) OR duration(i-1) >= 2 )”
or “(accentWeight >= 1 AND 2 < duration(i)) OR duration(i-1) >= 2”? When writing something that could be seen as arrogant such as “we leave that to the reader” then the reader needs to be given a fighting chance at figuring it out.

Discussion area is strong and well written.

After reading the paper I discovered the electronic Appendices, which give the paper a much more positive result for reproducability, etc. Very strong. Should be referenced in the text where they appear (the “Resulting rules.pdf” is especially interesting). The version of music21 used (and other software) should be mentioned in the interest of reproducing.

Additional comments

Paper needs some minor revisions noted above before accepting, but it is very close and I anticipate that the next version will be ready for publication.

The research area is of interest and the research design is well done, and well written. Professional, solid, and will have impact.

I am choosing not to publish my review publicly, but authors may know my name: Michael Scott Cuthbert (MIT).

·

Basic reporting

Good, but I have some misgivings about music terminology and some unusually translated terms: I'll paste a few paragraph from my letter here:

This paper should be accepted, but there are several minor revisions that should be undertaken. There are three main areas in which I would encourage revisions include: 1) tweaking some of the language usage such that it conforms to English musical terminology, 2) adding some citations and engagement with similar and related research, 3) making a few aspects of the presentation more clear.
First, there are some terminological snafus in the article. The term “easy” beat is incorrect: I believe the author means “weak” beat. The phrase “without getting noticed much” in line 125 is awkward and conceptually suspect. I believe the author means that these dissonances are less structural insomuch as they are heard as elaborating the surrounding consonances. In lines 127-128, the author might cite Jeppesen (or even the venerable J.J. Fux, 1725) to illustrate these dissonances because there are, in fact, some disagreements about categories of dissonance in this musical style. (There’s no reason to get into this debate in this paper, and it can be sidestepped by simply citing these standard dissonances.) In line 165, I believe the author means “whole note” rather than “whole tone,” since they’re talking about duration at this point. Additionally, it should be recognized that this music was not in 4/2 in the original manuscripts, but rather was probably signed with either a half circle or a cut half circle to indicate a duple grouping of the constituent breves. In other words, the 4/2 barring – and therefore the music21 metric weights – are modern anachronisms. (This fact doesn’t change the study’s results: it just should be acknowledged.) On line 199, the author means “tonic” not “root” (chords have roots, pieces and keys have tonics). Additionally, on this front, the abstract needs a careful edit. As it currently stands, it is difficult to read and uses unidiomatic words and turns of phrase.

Experimental design

It's good.

Validity of the findings

I would encourage the author to connect the findings to other existing research. Again, here are the relevant paragraphs:

The paper cites no other work machine learning on musical corpora or automated harmonic analysis. To my mind, even though no one has done this particular approach to dissonance treatment, the author should still cite other research into modeling musical harmonies. I’ll include a short bibliography below, which could provide a springboard for a more comprehensive literature review. Additionally, the author argues that their contribution is most valuable to automated musical composition research. I’d argue that this modeling is also valuable to research into statistical learning. You could argue that this article models the experience of a learner who tries to construct a series of cognitive templates through trial and error until settling on generalizable rules. The author might consider adding some discussion of these implications.
A few other discussions could be tightened, clarified, or removed. First, Table 2 doesn’t do a lot of work and takes up a lot of space. To my eye, all the table illustrates is that there exists very little variation within the clustered rules. I think the table should basically be deleted and perhaps replaced with another sentence (in the Quantitative Evaluation section) describing this lack of variation and the caveats present in the table’s footnotes. If the author believes the table to be important enough to stay in the article, more discussion is necessary to indicate why. Additionally, Figure 5 should be talked through step-by-step from top to bottom (see lines 360-380-ish) rather than the piecemeal approach used now, and the value i on which the dissonance actually occurs should be explicitly stated (I believe the dissonance occurs on i, and not i+1 or i-1, but I had to think about that for a bit). Also, explanations of computational specifics were often not comprehensible to me, but that’s probably because I’m a music theorists and not a computer scientist– I’ll defer to the editors and other reviewers on the assessment of those topics.

---

## Round 0.2 · accepted · Accept

The new version has received favorable comments from the reviewers, and I believe that it is now in a condition to be accepted.

·

Basic reporting

My concerns have been addressed. I'm glad this paper is moving toward publication!

Experimental design

My concerns have been addressed

Validity of the findings

My concerns have been addressed

Additional comments

My concerns are addressed. I congratulate the authors on a job well done.

---

## Author Rebuttal · Round 0.2

# Response to Reviewers' Comments

We thank the reviewers for their very helpful and positive comments and their detailed feedback. Below are their comments (in bold) together with our responses.

## Reviewer 1 (Anonymous)

**Basic reporting**
**On English:**
**Some typos in the abstract: "we us all other dissonances" -> "we use all other dissonances" and some shorthand also that could be made clearer. Abstract ll 14-16 could be broken into multiple sentences; and "with each other and *with* manually programmed rules" 16-17: "We chose genetic programming for our machine learning technique" I'm not sure what "as positive examples" means in l. 23 – "as examples used to train the algorithm in a particular dissonance category"? This seems redundant. The next lines 24-25 seem more important: "We also trained the algorithm in many types of negative examples such as…" l. 27 "category *of* suspension"**

**The cumbersome reading of the abstract almost made me not agree to review the paper; I hope others don't decide not to read it for the same reason.**

The abstract has been revised.

**The introduction's writing and later is stronger and better edited. But there are still errors that make reading hard – careful rereading by a native speaker is crucial:**
**35: methods OF algorithmic composition**
**144-5: "the algorithm better leaves" -- "an algorithm that leaves a few complex dissonance categories undetected is stronger than one that wrongly marks notes…" or something like that?**
**181: "To be at the save side" – "To be on the safe side"**

These errors have been corrected, and further proofreading has been done.

**On Literature references, sufficient field background/context:**
**The relevant literature is addressed and cited well. However, the original encoders of the Palestrina corpus should be mentioned as well as the editions from which they are encoded. (There are errors in encoding and in parsing that are known to people involved in the original project and the music21 project; I doubt they are sufficient to affect the end results however)**

A footnote with the relevant information has been added.

**On Professional article structure, figures. Raw data shared.:**
**Raw data is shared -- in abundance. Version of software used should be noted. Some issues w/ figures (discussed below) but in general very professional.**

**Figure 1 includes in addition to "p.t." "n.t." and "s." also "p.t.s". This could be read as "p.t." followed by "s." but the presumably correct reading as multiple passing tones is not included**

Figure 1 has been edited to only include the abbreviations explicitly listed in its caption (i.e. "p.t.s" has been replaced by two "p.t." markers).

**On Clear defs of all terms, theorems and proofs:**
**Very clear in general.**

**A pass, but some revision needed**
**Experimental design**
**The research question is well defined, and how this work on genetic programming will advance the field of computational music research is clearly stated and well argued (well done!).**

**There are no ethical concerns raised by the paper.**

**38–42 are key and important parts of the paper (and might be highlighted in the abstract): rules that can be studied by humans are important for acceptance of this work in the discipline of music theory (as an interdisciplinary paper should aim for).**

The revised abstract now also mentions that the resulting rules are human-readable.

**Cuthbert/Ariza/Friedland 2011 p. 391 (http://ismir2011.ismir.net/papers/PS3-6.pdf), among others, discuss the importance of understanding how a decision was made in the context of classification of musical features. Very important and good point.**

**148 It is unclear what the custom algorithm (after chordification) does in cases where the voice leading of the chordified version is unclear.**

We further clarified this section by adding the following:
The result of the "chordification" process on its own would loose all voice leading information (e.g., where voice crossing happens), but our algorithm processes those chords and the original polyphonic score in parallel and therefore such information is not lost. The algorithm finds the score notes that form a given chord by accessing the notes with the same start time (music21 parameter offset) as a given chord.

**153-159: this seems a VERY smart way of working. Applause.**

**The description of handling of suspensions (160-162) is still unclear to me though. After chordification, the dissonant pitches would already be tied to a previous pitch.**

That paragraph has been extended and now reads as follows:
"Suspensions are treated with special care. Remember that the algorithm processes the "chordified" score and the actual score in parallel. As a preprocessing step, the algorithm merges tied score notes into single notes to simplify their handling.  When the algorithm then loops through the chords and finds a dissonant note that started before the currently tested chord, it splits that note into two tied notes, and only the second note starting with the current chord is marked as dissonant."

**163-166: There are some cases in the Palestrina corpus of incorrect clefs (namely treble8vb as treble, causing fifths to become fourths). The authors are cautioned to check their corpus manually against the original encodings; all of which are on IMSLP. The authors should also be sure to be on the latest release of music21, as prior versions did not necessarily parse properly the (partially incorrectly encoded) krn files where the number of voices in Agnus II is fewer than in Agnus I, but the original (usually five-voice) structure is restored in Agnus III. This can make for a large number of dissonances.**

We eyeballed the results of the automatic dissonance labelling, where dissonances in music notation are marked by x-shaped note heads (see the folder 2 Preprocessing Results that is part of the provided data) and we did not find any labelled dissonances that where unreasonable.

We unsuccessfully tried (for several hours) to get our code working with the latest music21 version, but unfortunately were not able to do so (likely due to some changes of the behaviour of music21 sites and/or groups).

For completeness, we also compared the Agnus kern files of the versions v.2.1.2 (which we used) and the current version using git tools. The 'parent' of the composition (kern reference code OPT) is slightly changed (now always starts with "Missa"), but otherwise we could not find any changes in the many Agnus movements that we checked, certainly not any changes in clefs or number of parts.

**211: were melodic intervals given in semitones, as diatonic steps, or (perhaps best) in Hewlett's base-40 interval encoding? This information is needed to replicate. (I'm guessing from Table 1 later that it's diatonic steps)**

Added to text: "(measured in semitones)".

**253: are random values that match the positive examples excluded from set 3? Or are they too few to care about? (ah, this is mentioned in 282-83. I think they should be removed).**

Yes, as explained in the text, removing them would be the cleaner option, but the method works well enough without this, as the probability of this happening is indeed small.

**267: again the interval should be described as diatonic or semitonal?**

We added "semitones".

**285–325: my knowledge of this part of the paper is low. It looks good to me, but please weigh other reviewers' comments more highly for this section (Genetic Programming)**

**317-18: "The fitness assigned … is 10 times its accuracy…plus 0.001 times the size of the tree" – should this be minus 0.001 times the size of the tree? It seems to reward bloated trees?**

This has been corrected.

**Validity of the findings**
**132 – unclear why the further dissonance categories are irrelevant; they certainly do occur in the Palestrina Mass corpus, and at this point in the paper it is hard for the reader to know why rejection by the algorithms makes evidence irrelevant. (ah, l. 168 gives the answer to what the corpus is; that such dissonances do not occur is possible, but not certain)**

That sentence has now been removed. Instead, the following paragraph was added to the section Evaluation of the Dissonance Detection Algorithm:

Due to these precautions (avoiding wrongly labelled dissonances), due to the corpus (selected Agnus movements) and the frequency of certain dissonance categories in that corpus, only a subset of the standard dissonance categories were detected and learnt as rules in this project (see table 1). This point is further discussed below in the section on the evaluation of the clustering.

**171-175: not enough information is given here to verify the authors' claim that the detection works rather well. Can this data be provided (or see next comment)**

Unfortunately, we have no automatic way to evaluate the quality of our dissonance detection, we can only evaluate it by eyeballing. In order to support our claim we therefore now added a footnote that invites the interested reader to confirm our findings in music notation, i.e., MusicXML files, where detected dissonances are marked by x-shaped note heads, which are provided with that data for this paper.
We also revised this subsection further.

**183–186: a reader would be interested in a list of these occurrences of dissonances that cannot be automatically classified by the custom algorithm. They would be of interest for music theorists and Renaissance scholars in addition to future researchers who may want to improve the algorithm.**
Unfortunately, we cannot provide a systematic list of cases of dissonances not labelled by the algorithm. The original submission already contained the whole explanation of which dissonances are not labelled: If the algorithm does not find tones that, when removed, leave a consonant chord, then no note is labelled as a dissonance. Further, excluding dissonances with the parameter *max_diss_dur* avoided a considerable number of complex cases and otherwise wrongly labelled notes.
So, instead we added a footnote pointing out how the interested reader can investigate the unlabelled dissonances in music notation.

**196–199 is the most problematic passage I have yet read: the "key" detection of a Palestrina mass movement is likely to be very faulty as the Krumhansl-Schmuckler key determination algorithm (which should get citation) and the Sapp weights were designed for tonal music of the common practice period (1650–1880) and not for this period. I believe I have seen, in other forums, the authors' search for an algorithm that would classify Renaissance mode better, and I believe they can say that no such algorithm yet exists, and they should be able to satisfy me that the KS algorithm is the best thing available. But the argument needs to be made.**

The relevant paragraph has been revised and a footnote added. The mentioned citation has been added.

**215 – what does it mean to normalize the pitch class of the dissonance? does rotating pitch classes so, say G = 0, affect the results?**

Added brief explanation:
 the "normalised" pitch class of the dissonance (i.e. 0 is the root etc.).

**221 – the total number of points in each cluster are noted below, but the total number of points recognized as outliers should be noted. (it's at 246)**

Indeed, it is noted there.

**228-233 is very well explained.**

**Figure 2 is hard to read – the blue bars could be made more prominent and the dots bigger as well. A note that each pink dot represents one run could be added.**

Figure and caption have been revised.

**The bigger issue though is that C5 and C6 are included in the diagrams but have not been explained in the text yet. What are they? [Ah! From the scripts I discover they are the cases in l. 249; this should be explained]**

The cluster names have now been inserted into the text.

**Figure 3 – why not plot #s 0 0 0 0, 1 1 1 1, etc. instead of abstract figures that I need to return to the caption often to read? The switch from 0-1 to 0-10 score is also confusing.**

This figure has been revised according to these suggestions.

**Figures 5 and 6 should explain again what cluster number they refer to – we're going back and forth between cluster numbers and verbal descriptions.**

This has been added.

**Returning to 293–300 in light of Figure 5–6: 295: is the "dissonant" note necessarily dissonant? l. 263 seems to imply it could be simply a middle note.**

The discussion of figure has been revised, and also the following sentence has been added: "Remember that each rule states conditions that must hold between three consecutive melodic notes if the middle note is a dissonance."

**Figure 5 line 1: I read the right side as duration(i+3) and was wondering why the duration of the note three notes later was being taken into account. More space or something to show that it is duration(i) + 3 would help.**

Slightly more space has been added in this and similar situations.

**It took me some time to note that "dissonance status" was NOT an input variable, and that we are only looking at linear (melodic) segments. If this is true, then the results are even stronger, but either way, this point should be made very clear in the article.**

To better clarify this important point from the beginning, the introduction to the methods section now also states the following: "Each rule states conditions that must hold between three consecutive melodic notes if the middle note is a dissonance."

**355 – it is unclear here and in table 2, what "can be" means – does this mean that "the generated rule/solution allows the note to be an eighth note" or "the received rule allows the note to be an eighth note but the generated solution does not"? The direction of the deviation is important.**

The footnotes of table 2 have been extended for clarity.

**The Qualitative evaluation (339-356) is quite cursory and there is not enough information given here to evaluate the accuracy of the results, I'm afraid.**

That section now points more clearly to table 2, which summarises the findings of the qualitative evaluation. Also, the caption of table 2 has been revised for clarity and the table has been brought closer to this section for better clarity.

**374 – I'm not sure what an "easy beat" is. Is this a mistranslation for "simple beat" (which would also need explanation), or a integer number beat? It is not an English language term in UK or US.**

Corrected throughout paper to weak beat (that was a misplaced Germanism...).

**381: "We leave that to the reader" – the reader does not know the grouping – does the ^ at the beginning of the line constrain the entire line or not? So are lines 2-3 to be read as "accentWeight >= 1 AND ( 2 < duration(i) OR duration(i-1) >= 2 )"**
**or "(accentWeight >= 1 AND 2 < duration(i)) OR duration(i-1) >= 2"? When writing something that could be seen as arrogant such as "we leave that to the reader" then the reader needs to be given a fighting chance at figuring it out.**

Parentheses have been inserted to make this unambiguous.

**Discussion area is strong and well written.**

**After reading the paper I discovered the electronic Appendices, which give the paper a much more positive result for reproducability, etc. Very strong. Should be referenced in the text where they appear (the "Resulting rules.pdf" is especially interesting). The version of music21 used (and other software) should be mentioned in the interest of reproducing.**

The references to the supplementary material have been inserted, and the version information was added to the relevant Readme file.

**Comments for the Author**
**Paper needs some minor revisions noted above before accepting, but it is very close and I anticipate that the next version will be ready for publication.**

**The research area is of interest and the research design is well done, and well written. Professional, solid, and will have impact.**

**I am choosing not to publish my review publicly, but authors may know my name: Michael Scott Cuthbert (MIT).**

## Reviewer 2 (Christopher White)
**Basic reporting**
**Good, but I have some misgivings about music terminology and some unusually translated terms: I'll paste a few paragraph from my letter here:**

**This paper should be accepted, but there are several minor revisions that should be undertaken. There are three main areas in which I would encourage revisions include: 1) tweaking some of the language usage such that it conforms to English musical terminology, 2) adding some citations and engagement with similar and related research, 3) making a few aspects of the presentation more clear.**

The paper has been revised with these goals in mind.

**First, there are some terminological snafus in the article. The term "easy" beat is incorrect: I believe the author means "weak" beat.**

Replaced throughout.

The phrase "without getting noticed much" in line 125 is awkward and conceptually suspect. I believe the author means that these dissonances are less structural insomuch as they are heard as elaborating the surrounding consonances.

Revised with Jeppesen terminology (dissonances as secondary vs primary phenomenon).

In lines 127-128, the author might cite Jeppesen (or even the venerable J.J. Fux, 1725) to illustrate these dissonances because there are, in fact, some disagreements about categories of dissonance in this musical style. (There's no reason to get into this debate in this paper, and it can be sidestepped by simply citing these standard dissonances.)

Jeppesen citation added (and also Fux a bit later).

In line 165, I believe the author means "whole note" rather than "whole tone," since they're talking about duration at this point.

Fixed.

Additionally, it should be recognized that this music was not in 4/2 in the original manuscripts, but rather was probably signed with either a half circle or a cut half circle to indicate a duple grouping of the constituent breves. In other words, the 4/2 barring – and therefore the music21 metric weights – are modern anachronisms. (This fact doesn't change the study's results: it just should be acknowledged.)

Revised:
All examples in that subcorpus happen to be in 4/2 "meter" (tempus imperfectum denoted with cut half circle)

On line 199, the author means "tonic" not "root" (chords have roots, pieces and keys have tonics).

Corrected.

Additionally, on this front, the abstract needs a careful edit. As it currently stands, it is difficult to read and uses unidiomatic words and turns of phrase.

The abstract has been revised.

**Experimental design**
It's good.
**Validity of the findings**
I would encourage the author to connect the findings to other existing research. Again, here are the relevant paragraphs:

The paper cites no other work machine learning on musical corpora or automated harmonic analysis. To my mind, even though no one has done this particular approach to dissonance treatment, the author should still cite other research into modeling musical harmonies. I'll include a short bibliography below, which could provide a springboard for a more comprehensive literature review.

We politely disagree with the reviewer on this point. The original paper did already include references to literature on automatic analysis of Renaissance music. It also cited various research using inductive logic programming or genetic programming for music, i.e. the machine learning approaches related to this research (i.e. where rules or symbolic computer programs are learnt). The huge field of modelling tonal harmony is only indirectly related to this research. So, instead we added several seminal papers on modelling counterpoint composition in the section on Dissonances in Palestrina's Music.

**Additionally, the author argues that their contribution is most valuable to automated musical composition research. I'd argue that this modeling is also valuable to research into statistical learning. You could argue that this article models the experience of a learner who tries to construct a series of cognitive templates through trial and error until settling on generalizable rules. The author might consider adding some discussion of these implications.**

Using genetic programming in this way (the broad area of symbolic regression) is a well-established field of research with applications in other domains, we have referenced the paper by Schmidt & Lipson (2009), as well as the book by Poli, Langdon, & McPhee. In this version, we have added some sentences and further references in the discussion section.

**A few other discussions could be tightened, clarified, or removed. First, Table 2 doesn't do a lot of work and takes up a lot of space. To my eye, all the table illustrates is that there exists very little variation within the clustered rules. I think the table should basically be deleted and perhaps replaced with another sentence (in the Quantitative Evaluation section) describing this lack of variation and the caveats present in the table's footnotes. If the author believes the table to be important enough to stay in the article, more discussion is necessary to indicate why.**

Table 2 summarises the results of the qualitative evaluation. To better clarify this, it has been positioned closer to that section, and its caption has been revised.

**Additionally, Figure 5 should be talked through step-by-step from top to bottom (see lines 360-380-ish) rather than the piecemeal approach used now,**

This section has been revised. Still not all expressions of Figure 5 are discussed, but the reason for this has been emphasised stronger: "Note that the learnt rules contain some bloat – expressions that irrelevant or redundant. In our analysis we will focus on the meaningful expressions of the rule."

**and the value i on which the dissonance actually occurs should be explicitly stated (I believe the dissonance occurs on i, and not i+1 or i-1, but I had to think about that for a bit).**

Added sentence (as part of the discussion of figure 5):
"The dissonance occurs on the note with the index i, as in duration_i."

**Also, explanations of computational specifics were often not comprehensible to me, but that's probably because I'm a music theorists and not a computer scientist– I'll defer to the editors and other reviewers on the assessment of those topics.**
* * *
*Additional comments from the review letter not already included above:*

**This article outlines a way of organically learning "rules" of dissonance treatment in**

Palestina using an evolutionary algorithm that hones its expectations by modifying its proposed rules to conform to an annotated dataset. The specific contribution of this article is to demonstrate how this particular kind of machine-learning technique can indeed learn robust compositional tendencies. That is: this article shows that compositional tendencies that are usually presented in the classroom as a priori "rules" can be derived from a corpus of annotations.

This paper should be accepted, but there are several minor revisions that should be undertaken. There are three main areas in which I would encourage revisions include: 1) tweaking some of the language usage such that it conforms to English musical terminology, 2) adding some citations and engagement with similar and related research, 3) making a few aspects of the presentation more clear.

[...]

Again, I think this paper is a valuable contribution to the field of computational musicology, and would be a great addition to the literature after another round of edits.

**Small edits:**
173: "instead a wrong note labeled" should be reworded or edited
181: "Safe" not "save"

This has been corrected.

223-227: This could be tightened. Spending a paragraph on how the author represented the data to themselves seems extraneous.

We feel we should leave this paragraph; it is intended to make our evaluation of intermediate results (qualitative evaluation of the automatic clustering results in music notation) more transparent and reproducible. To further clarify this goal, we also added a pointer to where readers can find these intermediate clustering results in music notation.

290: There are two minus signs, one for arithmetic and one for negation.
294: I don't know what a "constant" is here.

This has been corrected / explained in more detail.

Some selected citations about analyzing chords and chord sequences in tonal music, some of which deal with classifying, parsing, or reducing out dissonances:
Pardo, Bryan, and William P. Birmingham. 1999. "Automated Partitioning of Tonal Music." Technical report, Electrical Engineering and Computer Science Department. University of Michigan.
White and Quinn. 2018. "Chord Context and Harmonic Function in Tonal Music." Music Theory Spectrum.
Temperley, David. 1997. "An Algorithm for Harmonic Analysis." Music Perception 15 (1): 31–68.
Rohrmeier, Martin, and Ian Cross. 2008. "Statistical Properties of Tonal Harmony in Bach's Chorales." in Proceedings of the 10th International Conference on Music Perception and Cognition. Sapporo: ICMPC: 619–627.

Please see comment above.

Again, we thank both reviewers for their valuable comments.